# Identification of AGXT2, SHMT1, and ACO2 as important biomarkers of acute kidney injury by WGCNA

**Jinshuang Wei<sup>☯</sup>, Junlin Zhang<sup>☯</sup>, Junyu Wei, Miaoyue Hu, Xiuqi Chen, Xuankai Qin, Jie Chen, Fengying Lei\*, Yuanhan Qin[ORCID]\***

The First Affiliated Hospital of Guangxi Medical University, Nanning, Guangxi, China

☯ These authors contributed equally to this work.
\* qinyuanhan603@163.com (YQ); fenghuang0999@163.com (FL)

## Abstract

Acute kidney injury (AKI) is a serious and frequently observed disease associated with high morbidity and mortality. Weighted gene co-expression network analysis (WGCNA) is a research method that converts the relationship between tens of thousands of genes and phenotypes into the association between several gene sets and phenotypes. We screened potential target genes related to AKI through WGCNA to provide a reference for the diagnosis and treatment of AKI. Key biomolecules of AKI were investigated based on transcriptome analysis. RNA sequencing data from 39 kidney biopsy specimens of AKI patients and 9 normal subjects were downloaded from the GEO database. By WGCNA, the top 20% of mRNAs with the largest variance in the data matrix were used to construct a gene co-expression network with a p-value < 0.01 as a screening condition, showing that the blue module was most closely associated with AKI. Thirty-two candidate biomarker genes were screened according to the threshold values of $|MM| \geq 0.86$ and $|GS| \geq 0.4$, and PPI and enrichment analyses were performed. The top three genes with the most connected nodes, alanine—glyoxylate aminotransferase 2(AGXT2), serine hydroxymethyltransferase 1 (SHMT1) and aconitase 2(ACO2), were selected as the central genes based on the PPI network. A rat AKI model was constructed, and the mRNA and protein expression levels of the central genes in the model and control groups were verified by PCR and immunohistochemistry experiments. The results showed that the relative mRNA expression and protein levels of AGXT2, SHMT1 and ACO2 showed a decrease in the model group. In conclusion, we inferred that there is a close association between AGXT2, SHMT1 and ACO2 genes and the development of AKI, and the down-regulation of their expression levels may induce AKI.

## Introduction

AKI is a serious and common disease with high morbidity and mortality [1]. It is a syndrome characterized by rapid loss of renal excretion. It is usually diagnosed by the accumulation of end products of nitrogen metabolism (urea and creatinine) and/or progressive reduction of

**Data Availability Statement:** The datasets generated and analyzed during the current study are available in the NCBI Gene Expression Omnibus repository (https://www.ncbi.nlm.nih.gov/geo/

query/acc.cgi?acc=GSE139061) with accession number GSE139061.

**Funding:** The author(s) received no specific funding for this work.

**Competing interests:** The authors have declared that no competing interests exist.

urine volume [2]. Due to the complexity of the pathophysiological process, the molecular mechanisms of acute kidney injury are not clearly defined. It has been suggested that the use of antiviral drugs causes drug-induced AKI, which may lead to AKI through mechanisms such as acute tubular necrosis (ATN), allergic interstitial nephritis (AIN), and crystal nephropathy [3]. Bioinformatics has been used for in silico analyses of biological queries using mathematical and statistical techniques [4].

WGCNA is a systems biology method for characterizing correlation patterns between genes in microarray samples. WGCNA can be used to find clusters (modules) of highly correlated genes, to aggregate such clusters using module signature gold or modular hub genes to correlate modules with each other (using external sample signature methods), and to calculate module membership measures. Correlation networks facilitate network-based genetic screening methods that can be used to identify candidate biomarkers or therapeutic targets [5]. WGCNA is now widely used in biological studies of diseases, physiology, drugs, evolution, and genomes [6], which identifies correlated gene clusters (modules), has been applied to identify biomarkers for Diabetic kidney disease [7] and Chronic Kidney Disease [8], demonstrating the feasibility of its use to identify biomarkers for AKI. AKI biomarkers are potential targets for AKI risk assessment, and with the advancement of genomics, proteomics and metabolomics research, new AKI markers can be explored; biological targets for AKI can be established through artificial intelligence learning to provide an experimental basis for clinical application and effectively reduce AKI rates.

## Materials and methods

### Data source and pre-processing

The GSE139061 dataset; containing 39 native human renal biopsy samples and 9 reference nephrectomies was downloaded from GEO database [1]. The Robust Multichip Average (RMA) method [9] in R software was then used to pre-process the downloaded raw data, including background correction, quantile correction normalization and expression integration. After removing outliers, there were 35 native human renal biopsy samples and 9 reference nephrectomies. The probes were annotated by annotation files, and duplicate genes were removed.

### WGCNA

The top 20% of genes with the highest variance were selected (4028 genes) for WGCNA analysis. A scale-free co-expression network was built using WGCNA package in R software [10]. In this study, the value of β was set to 14 (scale-free $R^2$ = 0.95) to ensure a scale-free network [10]. Next, the adjacency matrix was converted to a topological overlapmatrix (TOM) to cluster genes with similar expression profiles into modules using a mean linkage hierarchical clustering approach [11]. Notably, the minimum number of genes per gene network module was set to 30, and the Dynamic cut tree method algorithm was used to determine the gene network modules.

### Identification of candidate biomarkers

Gene connectivity was measured by the absolute value of the module membership (MM) score, which represents the Pearson correlation coefficient between a particular gene and the module trait value. We selected modules that met a p-value less than 0.001 for gene modules and clinical traits and calculated the gene significance (GS) scores, which represent the correlation between genes in these modules and each phenotype, were calculated in absolute values.

Candidate genes were screened by module membership (MM) scores and gene significance (GS) scores. $|MM| \geq 0.80$ indicates that the gene is correlated with the module, while $|GS| \geq 0.2$ requires that the gene expression profile is also correlated with the phenotype [5]. The module with the smallest p-value and the gene module that matched the clinical trait with a p-value of 0.0002 was selected, and candidate genes were screened using the following parameters, module significance$|MM| \geq 0.86$ and gene significance$|GS| \geq 0.4$.

## Co-expression network analysis and functional enrichment analysis

Based on the protein-protein interactions (PPI) from STRING (https://cn.string-db.org), PPI networks of candidate biomarker genes were constructed in each clinically significant module and visualized using Cytoscape. In addition, Gene ontology (GO) enrichment analysis and Kyoto encyclopedia of genes and genomes (KEGG) pathway analysis were performed using R package clusterProfiler [12, 13].

## Animal AKI model construction

Twelve male SD rats with 180–200 g were purchased from and housed in the Experimental Animal Center of Guangxi Medical University [Animal Certificate No.: SYXK Gui 2020–0004]. After one week of acclimatization feeding, AKI model was constructed by two consecutive tail vein injections of 5 mg/kg adriamycin. The animals were placed in a $CO_2$ anesthesia chamber, the $CO_2$ valve was opened, and after the animals were unconscious, without pinch reflex and without corneal reflex, the ventilation was continued for 2 minutes, and the blood and kidneys were taken after the experimental animals died. The experimental procedures were performed according to the review standards of the Animal Ethics Committee of Guangxi Medical University.

## Histopathology and immunohistochemistry

Seven days after the tail vein injection of adriamycin, blood test creatinine and urea nitrogen, kidneys were harvested and fixed in 10% formalin, embedded in paraffin, cut into thin sections and then HE stained to observe the kidney tissue damage. The paraffin sections were dewaxed, subjected to EDTA Antigen Retrieval Solution (pH 9.0), endogenous peroxidase blockage, serum closure at room temperature for 30 min. Next the sections were incubated with a primary rabbit anti-AGXT2 [14] (Sigma-Aldrich HPA037382), SHMT1 [15] (Abcam, ab186130) and ACO2 [16] (Rosemont, 11134-1-AP) antibody diluted 1:50 in 1% BSA incubation at 4˚C overnight, secondary antibody incubation at room temperature for 50 min, and DAB color development to quantitatively evaluate hub gene protein expression in the kidney tissue of AKI model and control mice.

## RNA isolation and RT-qPCR

Kidney tissue from control and AKI model rats, and total tissue RNA was extracted using a FastPure Cell RNA Isolation Kit V2 (Vazyme Cat. RC112.01) kit, and reverse transcribed using Vazyme's HiScript III RT SuperMix for qPCR (+gDNA wiper) The instructions were used to reverse transcribe the total RNA into cDNA and amplify it on a PCR amplifier. The primers were designed and synthesized by Bioengineering Co., Ltd(Shanghai), and the sequences are shown in Table 1. The qPCR mix contained 0.4 μL of each primer, 10.0 μL of master mix, 2.0 μL of cDNA, and ddH$_2$O to a final volume of 20.0 μL. The cycling conditions were as follows: 95˚C for 30 s, 1 cycle; 95˚C for 10s, 60˚C for 30 s, 40 cycles. The reaction

**Table 1. PCR primer sequences.**

| Primers | Sequence (5'-3') | | Product length/bp |
|---|---|---|---|
| β-actin | Forward | TGTCACCAACTGGGACGATA | 165 |
| | Reverse | GGGGTGTTGAAGGTCTCAAA | |
| ACO2 | Forward | GTGGGTGGTGATTGGAGATGAGAAC | 110 |
| | Reverse | TTGCGAAGCTCTTGGTGATGATGG | |
| AGXT2 | Forward | GCAGCAGTTGTGACCACTCCAG | 116 |
| | Reverse | ACCTCAAGCACAGCAGATCCAATG | |
| SHMT1 | Forward | CAGTTGAGAAGTCCGATCCTGTGTC | 110 |
| | Reverse | GTTGCCCTGTGTCGTGGAGATTC | |

conditions were as follows: 95˚C for 15s, 60˚C for 60s, 95˚C for 15s, 1 cycle. The relative expression of target genes was calculated using the $2^{-\triangle\triangle CT}$ method.

## Results

### Clustering of co-expression modules eigengenes in AKI

Expression profiles of 44 samples from two different kidney tissue sources were included in the co-expression analysis, with 4028 genes subjected to WGCNA analysis. After excluding the abnormal samples, no discrete samples were found by clustering the samples (Fig 1A); and to

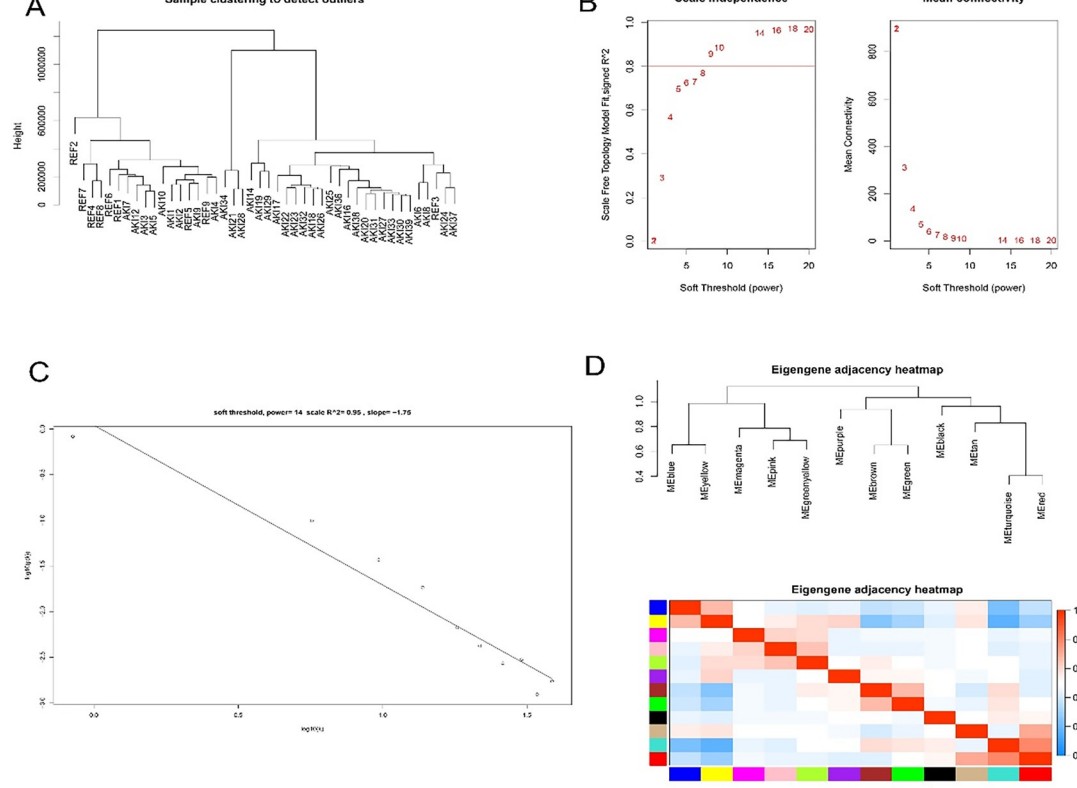

**Fig 1.** Determination of soft threshold parameters for WGCNA analysis: (A) analysis of outliers; (B) Analysis of scale-free fit indices at different soft threshold parameters β and determination of the average connectivity at soft threshold parameters; (C) Correlation of log (k) and log [P(k)]; (D) Sample clusters of module eigengenes.

ensure that the network was scale-free, an empirical analysis was performed to select the optimal parameter β. As depicted in Fig 1B and 1C, the scale-free topological model fit index (R2) and the average connectivity reached a steady state when β was equal to 14. Fig 1D displays the clusters of module eigengenes.

## Identification of key modules of AKI

After determining the weighting coefficients, the disTOM of 4028 genes was obtained (Fig 2A), and 13 modules were identified by mean linkage hierarchical clustering, each represented by a different color (Fig 2B). A heat-map was plotted to explore the association between module eigenvalues and AKI, as presented in Fig 2C, with each column showing the correlation coefficient and the corresponding p-value. Red represents positive correlations, and blue represents negative correlations, and the darker the color, the larger the correlation coefficient.

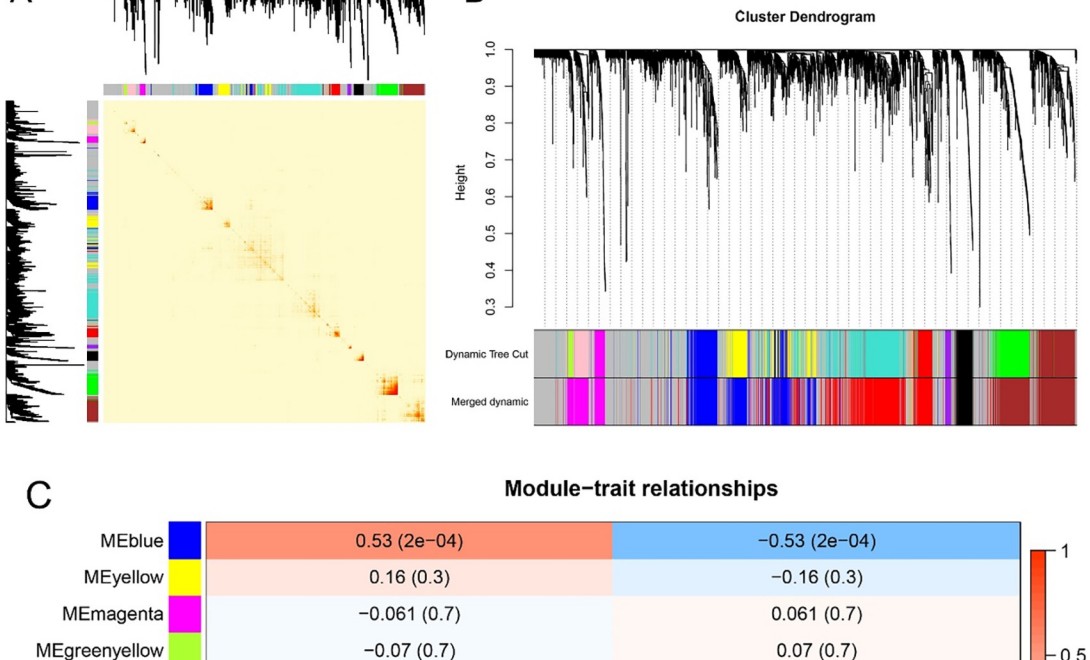

**Fig 2.** (A) Heat map of gene network visualization, the darker the red, the better the overlap; (B) Tree diagram of all differentially expressed genes based on clustering by the degree of difference (1-TOM); (C) Heat map of correlation between different modules and AKI.

AKI correlated most with the blue module, with a coefficient of 0.53 and p-value 0.0002. Therefore, it was selected as the module of significance for further analysis.

## Screening of candidate genes significantly associated with AKI

As described in Fig 3A, candidate biomarker genes were selected based on the thresholds |MM|≥0.86 and |GS|≥0.4 to obtain 32 candidate genes. As illustrated in Fig 3B, PPI networks covering these candidate genes in each module were constructed using Cytoscape based on PPI interactions from STRING, and the top three genes were selected as the final candidates, namely AGXT2, SHMT1, and ACO2.

## Hub gene enrichment analysis

GO, and KEGG analyses were performed to reveal the role of candidate genes in the pathogenesis of AKI. GO analysis (Fig 4A) revealed that candidate biomarker genes in biological processes were mainly enriched in the alpha-amino acid biosynthetic process [17–19] and the carboxylic acid biosynthetic process [20]; The molecular functions were mainly enriched in vitamin B and vitamin binding [21, 22], amino acid binding [23, 24] S-methyltransferase activity [25, 26]; The cellular components were significantly enriched in brush border [27, 28], apical plasma membrane [29, 30], apical part of cell [31]. In addition, as shown in Fig 4B, KEGG analysis indicated that candidate genes were mainly enriched in Glycine, serine and threonine metabolism [32, 33] Cysteine and methionine metabolism [34] and Biosynthesis of amino acids [35, 36]. The pathway enrichment analysis demonstrated that candidate biomarkers are mainly linked to amino acid synthesis and metabolism. Abnormal functions of these genes may cause abnormalities in amino acid synthesis and metabolism in the body and promote the development of kidney diseases.

## Experimental validation of the hub gene in a rat model of AKI

As seen in Table 2, serum creatinine and urea nitrogen increased, indicating that the AKI model was successfully constructed. Histopathologically, renal tissue presented significant cytoplasmic swelling and nuclear cleavage of tubular epithelial cells, and renal tubular cell extranuclear changes, mainly in the proximal tubules, as displayed in Fig 5A. The qPCR

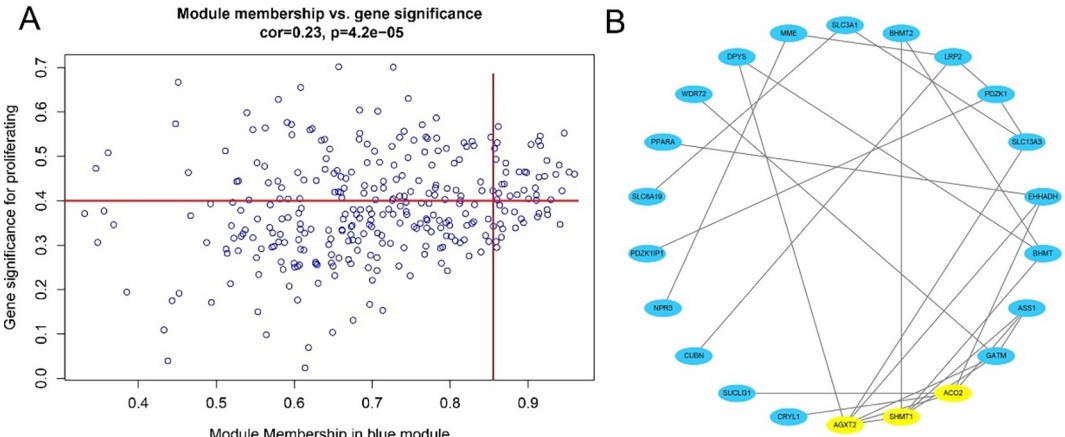

**Fig 3.** (A) Scatter plot of data concentration and P-value Cox regression, the X-axis indicates regression degree, the Y-axis indicates gene significance, and each circle represents a gene; (B) Network plot of key genes in the blue module, nodes indicate genes.

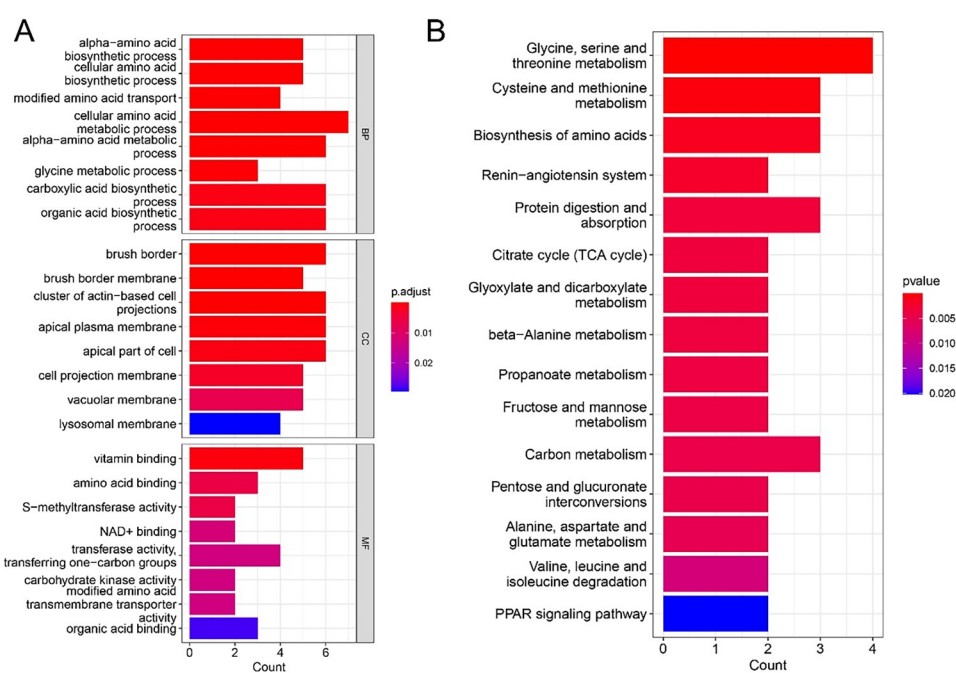

**Fig 4. GO and KEGG pathway enrichment analysis.** (A) GO analysis; (B) KEGG pathway analysis.

analysis (Fig 5B, Table 3) showed that the relative mRNA expression of AGXT2, SHMT1 and ACO2 in renal tissue of AKI rats decreased (p < 0.001). IHC staining revealed that AGXT2 and ACO2 were mainly expressed in the renal tubules, and SHMT1 was mainly expressed in glomeruli and blood vessels (Fig 5C, Table 4).

## Discussion

AKI is one of the most complex kidney diseases, and its molecular mechanisms have not been fully elucidated. Therefore, studies are urgently needed to identify potential biomarkers of AKI and to reveal the importance of molecular mechanisms for clinical practice. In this study, we analyzed the gene expression profiles of 35 AKI kidney biopsy samples and 9 normal kidneys and subjected 4028 genes to WGCNA analysis, identifying three genes, AGXT2, SHMT1 and ACO2, as hub genes. An AKI animal model was then constructed to verify the expression of these three genes, showing that AGXT2, SHMT1 and ACO2 were down-regulated in the kidney tissue of the model group.

Asymmetric dimethylarginine (ADMA), a metabolite of the amino acid L-arginine, competitively inhibits the enzymatic response to the cellular signal substance nitric oxide. AGXT2 metabolizes ADMA to ADGV (asymmetricα-keto-dimethylguanidinovaleric acid) which later can be excreted [37, 38] and decreased AGXT2 enzyme activity causes ADMA accumulation, leading to renal disease development. Low AGXT2 expression and activity have also been reported to affect symmetrical dimethylarginine (SDMA) metabolism, SDMA can compete for

**Table 2. Expression of serum creatinine and urea nitrogen.**

| Item | Control | AKI | T-test | p |
|---|---|---|---|---|
| blood serum creatinine | 19.20±1.83 | 78.56±45.16 | -3.179 | 0.024 |
| blood urea nitrogen | 8.91±0.68 | 22.92±0.78 | -33.108 | <0.001 |

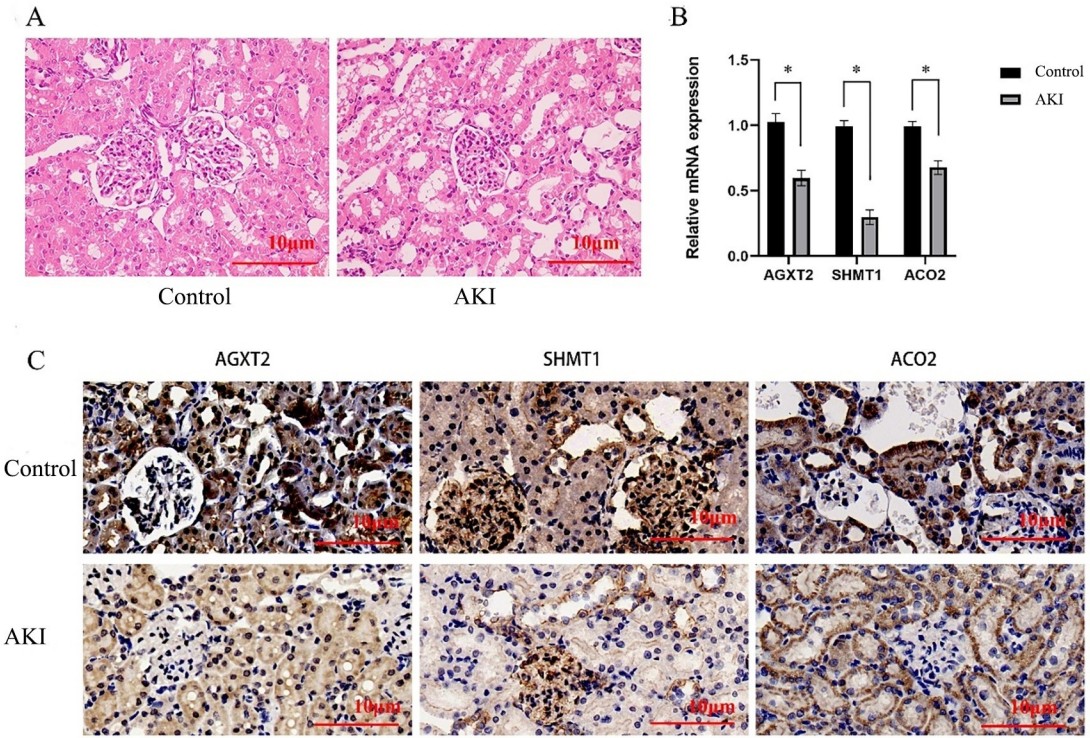

**Fig 5. Validation of hub gene expression in the rat model of AKI.** (A) HE staining of kidneys from control and AKI rats, magnification 40x. (B) Relative mRNA expression of hub genes in kidneys from control and AKI rats, *P<0.001. (C) IHC staining of the kidneys of control and AKI rats, magnification 40x.

**Table 3. Relative mRNA expression of the three genes.**

| Gene | Control | AKI | T-test | p |
|------|---------|-----|--------|---|
| AGXT2 | 1.02±0.06 | 0.60±0.06 | 9.606 | <0.001 |
| SHMT1 | 0.99±0.04 | 0.30±0.05 | 20.282 | <0.001 |
| ACO2 | 0.99±0.04 | 0.68±0.05 | 9.338 | <0.001 |

**Table 4. Expression of three proteins in renal tissue.**

| Gene | Control | AKI | T-test | p |
|------|---------|-----|--------|---|
| AGXT2 | 0.38±0.01 | 0.19±0.03 | 10.921 | <0.001 |
| SHMT1 | 0.31±0.04 | 0.18±0.01 | 5.404 | 0.006 |
| ACO2 | 0.33±0.03 | 0.24±0.03 | 3.810 | 0.019 |

L-arginine transport on membrane and indirectly inhibit NO synthesis, leading to renal dysfunction [39]. The reduced AGXT2 expression in AKI model rats validates this claim.

SHMT1 is a key enzyme in folate metabolism, providing the essential one-carbon unit for biosynthesis [40]. SHMT1 gene hypermethylation causes impaired folate metabolism and abnormal homocysteine (Hcy) remethylation through reduced SHMT1 expression, resulting in Hcy accumulation in the blood and hyperhomocysteinemia, which enhances oxidative stress in the kidney and induces structural damage and apoptosis in the podocytes, causing

kidney injury [41]. Silent expression of SHMT1 promotes bone-chondrogenic signaling in vascular smooth muscle cells by inducing cellular oxidative stress, thereby exacerbating phosphate-induced calcification of vascular smooth muscle cells and promoting chronic kidney injury [42], consistent with our experimental validation.

The mitochondrial ACO2 gene encodes an enzyme that catalyzes citrate conversion to isocitrate in the tricarboxylic acid cycle. Biallelic variants in ACO2 are purported to cause two distinct disorders: infantile cerebellar-retinal degeneration (ICRD), characterized by CNS abnormalities, neurodevelopmental phenotypes, optic atrophy, and retinal degeneration [43]. It has been reported that the expression of the aerobic oxidative metabolism-related protein ACO2 is reduced in urinary exosomes of diabetic patients [44]. downregulation of ACO2 expression activates ROS and induces HEK293T cell death [44, 45]. In the current study, the effect of ACO2 in the kidney is still underreported. Our animal model verified that ACO2 is lowly expressed in renal injury. The biomarkers we screened have only been validated on rat models and have not yet been involved in human experiments.

Our experimental results are based on animal models and have not been verified in patients with acute kidney injury. Patients will be collected for subsequent verification. The biomarkers we screened provide reference for the diagnosis and treatment of AKI.

## Conclusion

In clinical diagnosis and treatment, renal biopsy is used to extract renal tissue for rapid PCR detection of AGXT2, SHMT1 and ACO2, which has reached the possibility of evaluating AKI in patients.

## Acknowledgments

We thank all authors for their contributions to this research.

## Author Contributions

**Conceptualization:** Junyu Wei.

**Data curation:** Junyu Wei.

**Formal analysis:** Miaoyue Hu.

**Funding acquisition:** Miaoyue Hu.

**Investigation:** Xiuqi Chen.

**Methodology:** Xiuqi Chen.

**Project administration:** Xuankai Qin.

**Resources:** Xuankai Qin.

**Software:** Jie Chen.

**Supervision:** Jie Chen.

**Validation:** Fengying Lei.

**Visualization:** Fengying Lei.

**Writing – original draft:** Jinshuang Wei, Junlin Zhang.

**Writing – review & editing:** Jinshuang Wei, Junlin Zhang, Yuanhan Qin.

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
