## [Decision Letter · Decision Letter 0]

17 Nov 2022

PONE-D-22-24410Identification of AGXT2, SHMT1, and ACO2 as important biomarkers of AKI by WGCNAPLOS ONE

Dear Dr. qin,

Thank you for submitting your manuscript to PLOS ONE. After careful consideration, we feel that it has merit but does not fully meet PLOS ONE’s publication criteria as it currently stands. Therefore, we invite you to submit a revised version of the manuscript that addresses the points raised during the review process.

We look forward to receiving your revised manuscript.

Kind regards,

Utpal Sen, Ph.D.

Academic Editor

PLOS ONE

Journal Requirements:

https://pubmed.ncbi.nlm.nih.gov/32449285/

In your revision ensure you cite all your sources (including your own works), and quote or rephrase any duplicated text outside the methods section. Further consideration is dependent on these concerns being addressed.

3. To comply with PLOS ONE submissions requirements, in your Methods section, please provide additional information regarding the experiments involving animals and ensure you have included details on (1) methods of sacrifice, (2) methods of anesthesia and/or analgesia, and (3) efforts to alleviate suffering.

This work was financially supported by a grant from Guangxi Medical High-level Key Talents /139/ Project (No.G201901010), Guangxi Natural Science Fund Youth Science Foundation Project (No. 2018GXNSFBA050014).

Reviewers' comments:

Reviewer's Responses to Questions

**Comments to the Author**

1. Is the manuscript technically sound, and do the data support the conclusions?

Reviewer #1: Partly

Reviewer #2: Yes

2. Has the statistical analysis been performed appropriately and rigorously? 

Reviewer #1: I Don't Know

Reviewer #2: Yes

3. Have the authors made all data underlying the findings in their manuscript fully available?

Reviewer #1: Yes

Reviewer #2: Yes

4. Is the manuscript presented in an intelligible fashion and written in standard English?

Reviewer #1: No

Reviewer #2: Yes

5. Review Comments to the Author

Reviewer #1: Acute kidney injury (AKI) is a rapidly growing health burden. The complexity of pathogenesis keeps the molecular mechanisms not fully investigated. Wei and colleagues performed a screening analysis of RNA sequencing data from kidney biopsy specimens from healthy donors and patients with AKI. Analysis has been done with bioinformatics methods to search for genes that are associated with kidney injury. Computational analyses have been complemented by animal experiments.

The computational part meets no comments and looks well designed and performed.

There are a few major and minor comments on animal experiments:

Major comment:

1. The authors selected the adriamycin model of kidney injury which is commonly used for studying chronic kidney injury rather than acute form (PMID: 21175974). In this case, the selected animal model is not giving a valid reason to compare data from humans and rats and moreover, to use data from rats to validate data generated from patience.

However, the description of the design of the animal experiment is quite superficial. If there is some model optimization, that makes it suitable for research in AKI field, this data must be placed in the method section.

Minor comments:

1. IHC staining has been performed to show protein expression levels. Imaging of the performed staining was done with 10x magnification which is good for overview. At the same time, authors specify structure-specific expression patterns. For this reason, I would highly recommend imaging with a higher magnification (40x). Please, add scale bars to all pictures. Also, the list of antibodies (with ref.) used for staining should be added to the method section.

2. Please, do a revision of the whole document, there is an inconsistency in formatting (2.2, 2.6, Figure 5).

3. Please, add a section with a statistical analysis of data from the animal experiment. Fig. 5 B. Please, depict bars with “*” where you observe the significant difference.

4. Introduction: “Therefore, we investigated the key genes that cause AKI based on transcriptome profiling using RNA sequencing data from kidney biopsy specimens of 39 AKI patients and 9 normal subjects from GEO database”. // I would not say that selected genes cause AKI but rather are associated.

5. Results (3.4). “The molecular functions are mainly enriched in vitamin B and vitamin B. The molecular functions were mainly enriched in vitamin binding[14, 15] , amino acid binding[16, 17] S-methyltransferase activity[18, 19] The molecular functions are mainly enriched in vitamin binding, amino acid binding, and S-methyltransferase activity.” // Multiple self-repeating, please rewrite this section.

6. Discussion: “Asymmetric dimethylarginine (ADMA), a metabolite of the amino acid L-arginine, competitively inhibits the enzymatic response to the cellular signal substance nitric oxide. AGXT2 enzyme in the kidney promotes ADMA excretion…”. // AGXT2 is not promoting ADMA excretion. AGXT2 metabolizes ADMA to ADGV (asymmetric α-keto-dimethylguanidinovaleric acid) which later can be excreted (PMID: 31533264).

7. Discussion: “SHMT1 gene hypermethylation causes impaired folate metabolism and abnormal homocysteine (Hcy) remethylation through reduced SHMT1 expression, resulting in Hcy accumulation in the blood and hyperhomocysteinemia, which enhances oxidative stress in the kidney and induces structural damage and apoptosis in the pedal cells, causing kidney injury”. // I guess the authors mean podocytes.

Reviewer #2: Title; Identification of AGXT2, SHMT1, and ACO2 as important biomarkers of AKI by WGCNA

Comments; In my view, the results obtained in this study are worthy for publication. The manuscript needs major essential revision before publication. I would like to overview the revised version of the manuscript. I have the following comments/suggestions for authors to address before final decision on the manuscript.

1. The abstract background section lacks rational and objectives of the study. In addition the method section is also written incorrectly.

2. Authors have written “Thirty-two candidate biomarker genes were screened based on the thresholds |MM|≥0.86 and |GS|≥0.4”. Justify the basis of threshold value in the manuscript with relevant references.

3. Authors have to elaborate the conclusion part of the abstract based on findings.

4. Authors have not provided the data regarding selection of only 39 patients. Are there only 39 patients of AKI, then what was the need of this analysis.

5. Authors have to add a summary in the conclusion section on: How can we translate molecular patterns of injury into diagnostic tests?

6. The kidney is a sensor of the environment and an effector of homeostasis then how the data of this manuscript will reflect the central notion in Nephrology.

7. In the Introduction section the author should refer to the research paper and comment on recent in-silico techniques. It will be good information for the readers. I would like to recommend several papers, among many others, providing further explanation on this topic:PMID: 21989830 PMID: 23383724 PMID: 35315127 PMID: 35362492 PMID: 35276295 PMID: 28681927 PMID: 35486518 PMID: 27699663 PMID: 27194485 PMID: 36307910 PMID: 36336960

8. Introduction section lacks a proper description of the present study and on its background. Write two to three previous studies in connection to the present study with reference. Define the proper objective of your study, as it is not clearly mentioned in the abstract section too.

9. In the introduction section authors have written about methods in place of brief description of AKI.

10. In methodology, as authors have said that they have downloaded 9 normal subjects from the GEO database. Was there any specific reason to choose only 9subjects or do you use any other filter to choose them?

11. Rephrase the sentence “We used a robust Robust Multichip Average (RMA) method in R software” by removing the repetition of the words and also provide the reference.

12. Authors wrote, “In this study, the value of β was set to 14 (scale-free R2 = 0.95) to ensure a scale-free network”. How the value of β was set give the reference.

13. “As mentioned in the Figure 3A, candidate biomarker genes were selected based on the thresholds |MM|≥0.86 and |GS|≥0.4 to obtain 32 candidate genes. How did the authors know that they get 32 candidate genes in accordance with the written statement? Correct the controversial statement.

14. The authors have used too many undefined abbreviations in the title of the manuscript. Re-phrase/re-write accordingly.

15. Define non-standard abbreviations in the Abstract.

16. Clearly define the aim and objectives of the study in the last paragraph of the Introduction section.

17. The references in the Methods section should be placed next to the relevant information instead next to the subheadings.

18. Present the limitations of the study in the last paragraph of the Discussions section.

19. A short conclusions section could be added at the end of the Discussions section.

20. “It has been suggested that the use of antiviral drugs causes drug-induced acute kidney injury,” Authors should use AKI here instead of full form.

21. “We used a robust Robust Multichip Average (RMA) method in R software,” Repetition of words in the line.

22. “Tab.1 PCR primer sequences” Mention the table name appropriately. Whoes length/bp is given column length/bp in the table.

23. “The molecular functions are mainly enriched in vitamin B and vitamin B.” Did not understand what the authors were trying to say.

24. “The molecular functions were mainly enriched in vitamin binding [14, 15] , amino acid binding [16, 17] S-methyltransferase activity [18, 19] The molecular functions are mainly enriched in vitamin binding, amino acid binding, S-methyltransferase activity.” Repetition of line. Authors should clearly mention whose molecular function they are talking about.

25. Throughout the MS authors have written “Tab” instead of table. Authors need to correct it.

26. “Tab.4 Expression of three proteins in renal tissue” What about the units? Is there no need to mention units with the values in the table for protein expression?

27. “The current study found that ACO2 mainly affects neurological and ophthalmic disorders.” The focus of the current study is AKI then how authors justify the statement written here about ACO2.

6. PLOS authors have the option to publish the peer review history of their article (what does this mean?). If published, this will include your full peer review and any attached files.

Reviewer #1: No

Reviewer #2: No

---

## [Author Response · Author response to Decision Letter 0]

25 Nov 2022

Response to Reviews

First of all, I would like to thank the editor and the reviewers for their recognition of my manuscript, and I have revised the manuscript accordingly, and here are the answers to the questions raised by the reviewers.

Reviewer #1: Acute kidney injury (AKI) is a rapidly growing health burden. The complexity of pathogenesis keeps the molecular mechanisms not fully investigated. Wei and colleagues performed a screening analysis of RNA sequencing data from kidney biopsy specimens from healthy donors and patients with AKI. Analysis has been done with bioinformatics methods to search for genes that are associated with kidney injury. Computational analyses have been complemented by animal experiments.

The computational part meets no comments and looks well designed and performed.

I would like to thank reviewer 1 for his approval of my manuscript.

There are a few major and minor comments on animal experiments:

Major comment:

Question 1. The authors selected the adriamycin model of kidney injury which is commonly used for studying chronic kidney injury rather than acute form (PMID: 21175974). In this case, the selected animal model is not giving a valid reason to compare data from humans and rats and moreover, to use data from rats to validate data generated from patience.

However, the description of the design of the animal experiment is quite superficial. If there is some model optimization, that makes it suitable for research in AKI field, this data must be placed in the method section.

Answer 1. Thank you for your pointers. After reviewing the literature and studying it, we found the following literature is describing the acute kidney injury of Adriamycin(Prevention and possible mechanism of a purified Laminaria japonica polysaccharide on adriamycin-induced acute kidney injury in mice PMID:31958563

Podocyte protection by Angptl3 knockout via inhibiting ROS/GRP78 pathway in LPS-induced acute kidney injury PMID:35086056

Nephroprotective mechanisms of Ambrette (Abelmoschus moschatus Medik.) leaf extracts in adriamycin mediated acute kidney injury model of Wistar rats PMID:35339624

Transplantation of mesenchymal stem cells preserves podocyte homeostasis through modulation of parietal epithelial cell activation in adriamycin-induced mouse kidney injury model PMID:33124682）.

Minor comments: 

Question 1. IHC staining has been performed to show protein expression levels. Imaging of the performed staining was done with 10x magnification which is good for overview. At the same time, authors specify structure-specific expression patterns. For this reason, I would highly recommend imaging with a higher magnification (40x). Please, add scale bars to all pictures. Also, the list of antibodies (with ref.) used for staining should be added to the method section.

Answer 1. Thank you, we have revised in the manuscript.

Question 2. Please, do a revision of the whole document, there is an inconsistency in formatting (2.2, 2.6, Figure 5).

Answer 2. Thank you, we have revised in the manuscript. 

Question 3. Please, add a section with a statistical analysis of data from the animal experiment. Fig. 5 B. Please, depict bars with “*” where you observe the significant difference.

Answer 3. Thank you, we have revised in the manuscript.

Question 4. Introduction: “Therefore, we investigated the key genes that cause AKI based on transcriptome profiling using RNA sequencing data from kidney biopsy specimens of 39 AKI patients and 9 normal subjects from GEO database”. // I would not say that selected genes cause AKI but rather are associated.

Answer 4. Thank you, we have revised in the manuscript.

Question 5. Results (3.4). “The molecular functions are mainly enriched in vitamin B and vitamin B. The molecular functions were mainly enriched in vitamin binding[14, 15] , amino acid binding[16, 17] S-methyltransferase activity[18, 19] The molecular functions are mainly enriched in vitamin binding, amino acid binding, and S-methyltransferase activity.” // Multiple self-repeating, please rewrite this section.

Answer 5. Thank you, we have revised in the manuscript.

Question 6. Discussion: “Asymmetric dimethylarginine (ADMA), a metabolite of the amino acid L-arginine, competitively inhibits the enzymatic response to the cellular signal substance nitric oxide. AGXT2 enzyme in the kidney promotes ADMA excretion…”. // AGXT2 is not promoting ADMA excretion. AGXT2 metabolizes ADMA to ADGV (asymmetric α-keto-dimethylguanidinovaleric acid) which later can be excreted (PMID: 31533264).

Answer 6. Thank you, we have revised in the manuscript.

Question 7. Discussion: “SHMT1 gene hypermethylation causes impaired folate metabolism and abnormal homocysteine (Hcy) remethylation through reduced SHMT1 expression, resulting in Hcy accumulation in the blood and hyperhomocysteinemia, which enhances oxidative stress in the kidney and induces structural damage and apoptosis in the pedal cells, causing kidney injury”. // I guess the authors mean podocytes.

Answer 7. Thank you, we are indeed talking about podocytes.

Reviewer #2: Title; Identification of AGXT2, SHMT1, and ACO2 as important biomarkers of AKI by WGCNA

Comments; In my view, the results obtained in this study are worthy for publication. The manuscript needs major essential revision before publication. I would like to overview the revised version of the manuscript. I have the following comments/suggestions for authors to address before final decision on the manuscript.

I would like to thank reviewer 2 for his approval of my manuscript.

Question 1. The abstract background section lacks rational and objectives of the study. In addition the method section is also written incorrectly.

Answer 1. Thank you, we have revised in the manuscript.

Question 2. Authors have written “Thirty-two candidate biomarker genes were screened based on the thresholds |MM|≥0.86 and |GS|≥0.4”. Justify the basis of threshold value in the manuscript with relevant references.

Answer 2. Gene connectivity was measured by the absolute value of the module membership (MM) score, which represents the Pearson correlation coefficient between a particular gene and the module trait value. we selected modules that met a p-value less than 0.001 for gene modules and clinical traits and calculated the gene significance (GS) scores, which represent the correlation between genes in these modules and each phenotype, were calculated in absolute values. Candidate genes were screened by module membership (MM) scores and gene significance (GS) scores. |MM| ≥ 0.80 indicates that the gene is correlated with the module, while |GS| ≥ 0.2 requires that the gene expression profile is also correlated with the phenotype(PMID: 19114008)

Question 3. Authors have to elaborate the conclusion part of the abstract based on findings.

Answer 3. Thank you, we have revised in the manuscript.

Question 4. Authors have not provided the data regarding selection of only 39 patients. Are there only 39 patients of AKI, then what was the need of this analysis.

Answer 4. The GSE139061 dataset; only containing 39 native human renal biopsy samples and 9 reference nephrectomies was downloaded from GEO database 

Question 5. Authors have to add a summary in the conclusion section on: How can we translate molecular patterns of injury into diagnostic tests?

Answer 5. Thank you, we have revised in the manuscript.

Question 6. The kidney is a sensor of the environment and an effector of homeostasis then how the data of this manuscript will reflect the central notion in Nephrology.

Answer 6. We used the same batch of SD male rats with similar growth status, weighing 180-200 g, and housed in SPF class animal rooms to reduce the influence of the environment on the experiments.

Question 7. In the Introduction section the author should refer to the research paper and comment on recent in-silico techniques. It will be good information for the readers. I would like to recommend several papers, among many others, providing further explanation on this topic:PMID: 21989830 PMID: 23383724 PMID: 35315127 PMID: 35362492 PMID: 35276295 PMID: 28681927 PMID: 35486518 PMID: 27699663 PMID: 27194485 PMID: 36307910 PMID: 36336960

Answer 7. We have added some corresponding content in the introduction section.

Question 8. Introduction section lacks a proper description of the present study and on its background. Write two to three previous studies in connection to the present study with reference. Define the proper objective of your study, as it is not clearly mentioned in the abstract section too.

Answer 8. Thank you, we have revised in the manuscript.

Question 9. In the introduction section authors have written about methods in place of brief description of AKI.

Answer 9. Thank you, we have revised in the introduction.

Question 10. In methodology, as authors have said that they have downloaded 9 normal subjects from the GEO database. Was there any specific reason to choose only 9subjects or do you use any other filter to choose them?

Answer 10. The GSE139061 dataset; only containing 39 native human renal biopsy samples and 9 reference nephrectomies was downloaded from GEO database

Question 11. Rephrase the sentence “We used a robust Robust Multichip Average (RMA) method in R software” by removing the repetition of the words and also provide the reference.

Answer 11. Thank you, we have revised in the manuscript.

Question 12. Authors wrote, “In this study, the value of β was set to 14 (scale-free R2 = 0.95) to ensure a scale-free network”. How the value of β was set give the reference.

Answer 12. Thank you, we have revised in the manuscript.

Question 13. “As mentioned in the Figure 3A, candidate biomarker genes were selected based on the thresholds |MM|≥0.86 and |GS|≥0.4 to obtain 32 candidate genes. How did the authors know that they get 32 candidate genes in accordance with the written statement? Correct the controversial statement.

Answer 13. MM represents the horizontal coordinate, GS represents the vertical coordinate, and the upper right quadrant of the intersection of the two is the candidate gene.

Question 14. The authors have used too many undefined abbreviations in the title of the manuscript. Re-phrase/re-write accordingly.

Answer 14. Thank you, we have revised in the manuscript.

Question 15. Define non-standard abbreviations in the Abstract.

Answer 15. Thank you, we have revised in the manuscript.

Question 16. Clearly define the aim and objectives of the study in the last paragraph of the Introduction section.

Answer 16. Thank you, we have revised in the manuscript.

Question 17. The references in the Methods section should be placed next to the relevant information instead next to the subheadings.

Answer 17. Thank you, we have revised in the manuscript.

Question 18. Present the limitations of the study in the last paragraph of the Discussions section.

Answer 18. Thank you, we have revised in the manuscript.

Question 19. A short conclusions section could be added at the end of the Discussions section.

Answer 19. Thank you, we have revised in the manuscript.

Question 20. “It has been suggested that the use of antiviral drugs causes drug-induced acute kidney injury,” Authors should use AKI here instead of full form.

Answer 20. Thank you, we have revised in the manuscript.

Question 21. “We used a robust Robust Multichip Average (RMA) method in R software,” Repetition of words in the line.

Answer 21. Thank you, we have revised in the manuscript.

Question 22. “Tab.1 PCR primer sequences” Mention the table name appropriately. Whoes length/bp is given column length/bp in the table.

Answer 22. Thank you, we have revised in the manuscript.

Question 23. “The molecular functions are mainly enriched in vitamin B and vitamin B.” Did not understand what the authors were trying to say.

Answer 23. Thank you, we have revised in the manuscript.

Question 24. “The molecular functions were mainly enriched in vitamin binding [14, 15] , amino acid binding [16, 17] S-methyltransferase activity [18, 19] The molecular functions are mainly enriched in vitamin binding, amino acid binding, S-methyltransferase activity.” Repetition of line. Authors should clearly mention whose molecular function they are talking about.

Answer 24. Thank you, we have revised in the manuscript.

Question 25. Throughout the MS authors have written “Tab” instead of table. Authors need to correct it.

Answer 25. Thank you, we have revised in the manuscript.

Question 26. “Tab.4 Expression of three proteins in renal tissue” What about the units? Is there no need to mention units with the values in the table for protein expression?

Answer 26. This is the relative protein expression, no units.

Question 27. “The current study found that ACO2 mainly affects neurological and ophthalmic disorders.” The focus of the current study is AKI then how authors justify the statement written here about ACO2.

Answer 27. Thank you, we have revised in the manuscript.

---

## [Decision Letter · Decision Letter 1]

2 Jan 2023

PONE-D-22-24410R1Identification of AGXT2, SHMT1, and ACO2 as important biomarkers of acute kidney injury  by WGCNAPLOS ONE

Dear Dr. qin,

Thank you for submitting your manuscript to PLOS ONE. After careful consideration, we feel that it has merit but does not fully meet PLOS ONE’s publication criteria as it currently stands. Therefore, we invite you to submit a revised version of the manuscript that addresses the points raised during the review process.

We look forward to receiving your revised manuscript.

Kind regards,

Utpal Sen, Ph.D.

Academic Editor

PLOS ONE

Journal Requirements:

Reviewers' comments:

Reviewer's Responses to Questions

**Comments to the Author**

1. If the authors have adequately addressed your comments raised in a previous round of review and you feel that this manuscript is now acceptable for publication, you may indicate that here to bypass the “Comments to the Author” section, enter your conflict of interest statement in the “Confidential to Editor” section, and submit your "Accept" recommendation.

Reviewer #1: All comments have been addressed

Reviewer #2: All comments have been addressed

2. Is the manuscript technically sound, and do the data support the conclusions?

Reviewer #1: Yes

Reviewer #2: (No Response)

3. Has the statistical analysis been performed appropriately and rigorously? 

Reviewer #1: Yes

Reviewer #2: Yes

4. Have the authors made all data underlying the findings in their manuscript fully available?

Reviewer #1: Yes

Reviewer #2: Yes

5. Is the manuscript presented in an intelligible fashion and written in standard English?

Reviewer #1: Yes

Reviewer #2: Yes

6. Review Comments to the Author

Reviewer #1: I would like to thank authors for the addressing of comments from the initial revision.

Please find my additional recommendation below:

Question 1. The authors selected the adriamycin model of kidney injury which is commonly used for studying chronic kidney injury rather than acute form (PMID: 21175974). In this case, the selected animal model is not giving a valid reason to compare data from humans and rats and moreover, to use data from rats to validate data generated from patience. However, the description of the design of the animal experiment is quite superficial. If there is some model optimization, that makes it suitable for research in AKI field, this data must be placed in the method section. Answer 1. Thank you for your pointers. After reviewing the literature and studying it, we found the following literature is describing the acute kidney injury of Adriamycin(Prevention and possible mechanism of a purified Laminaria japonica polysaccharide on adriamycin induced acute kidney injury in mice PMID:31958563 Podocyte protection by Angptl3 knockout via inhibiting ROS/GRP78 pathway in LPS induced acute kidney injury PMID:35086056 Nephroprotective mechanisms of Ambrette (Abelmoschus moschatus Medik.) leaf extracts in adriamycin mediated acute kidney injury model of Wistar rats PMID:35339624 Transplantation of mesenchymal stem cells preserves podocyte homeostasis through modulation of parietal epithelial cell activation in adriamycin-induced mouse kidney injury model PMID:33124682.

All these references strongly prove the choice of the model. I would like to suggest citing them in the animal-related chapter of the material and method section.

Minor comments: Question 1. IHC staining has been performed to show protein expression levels. Imaging of the performed staining was done with 10x magnification which is good for overview. At the same time, authors specify structure-specific expression patterns. For this reason, I would highly recommend imaging with a higher magnification (40x). Please, add scale bars to all pictures. Also, the list of antibodies (with ref.) used for staining should be added to the method section. Answer 1. Thank you, we have revised in the manuscript.

Thank you for addressing this comment.

Question 2. Please, do a revision of the whole document, there is an inconsistency in formatting (2.2, 2.6, Figure 5). Answer 2. Thank you, we have revised in the manuscript.

Thank you for addressing this comment.

Question 3. Please, add a section with a statistical analysis of data from the animal experiment. Fig. 5 B. Please, depict bars with “*” where you observe the significant difference. Answer 3. Thank you, we have revised in the manuscript.

Thank you for addressing this comment.

Question 4. Introduction: “Therefore, we investigated the key genes that cause AKI based on transcriptome profiling using RNA sequencing data from kidney biopsy specimens of 39 AKI patients and 9 normal subjects from GEO database”. // I would not say that selected genes cause AKI but rather are associated. Answer 4. Thank you, we have revised in the manuscript.

Thank you for addressing this comment.

Question 5. Results (3.4). “The molecular functions are mainly enriched in vitamin B and vitamin B. The molecular functions were mainly enriched in vitamin binding[14, 15] , amino acid binding[16, 17] S-methyltransferase activity[18, 19] The molecular functions are mainly enriched in vitamin binding, amino acid binding, and S-methyltransferase activity.” // Multiple self-repeating, please rewrite this section. Answer 5. Thank you, we have revised in the manuscript.

Thank you for addressing this comment.

Question 6. Discussion: “Asymmetric dimethylarginine (ADMA), a metabolite of the amino acid L-arginine, competitively inhibits the enzymatic response to the cellular signal substance nitric oxide. AGXT2 enzyme in the kidney promotes ADMA excretion…”. // AGXT2 is not promoting ADMA excretion. AGXT2 metabolizes ADMA to ADGV (asymmetric α-keto dimethylguanidinovaleric acid) which later can be excreted (PMID: 31533264). Answer 6. Thank you, we have revised in the manuscript.

Thank you for addressing this comment.

Question 7. Discussion: “SHMT1 gene hypermethylation causes impaired folate metabolism and abnormal homocysteine (Hcy) remethylation through reduced SHMT1 expression, resulting in Hcy accumulation in the blood and hyperhomocysteinemia, which enhances oxidative stress in the kidney and induces structural damage and apoptosis in the pedal cells, causing kidney injury”. // I guess the authors mean podocytes. Answer 7. Thank you, we are indeed talking about podocytes.

Please, use podocytes instead of pedal cells.

I would like to leave a few additional comments:

“A rat AKI model was constructed, and the mRNA and protein expression levels of the central genes in the model and normal groups were verified by PCR and immunohistochemistry experiments”. Please, use word control instead of normal (in this sentence and further in the manuscript).

“Screening potential core genes related to AKI by WGCNA; to explore the potential biomolecules of AKI and provide a reference for the diagnosis and treatment of AKI.” // Difficult for understanding sentence, please rephrase it.

“…genes to correlate modules with each other (using external sample signature) methods), and to…”. // Please, remove extra bracket.

“… we selected modules that met a p-value less than 0.001…” // Please, start the sentence with a capital letter.

“antigen repair”. The right term for this procedure is retrieval. Please also specify which buffer was used for this procedure.

“Normal rats and AKI model rats were collected…”// Please exchange to “Kidney tissue from control and AKI model rats…”.

In the discussion section you are talking about SDMA and renal dysfunction. Could you please briefly specify the mechanism? (e.g. Competition for L-arginine transport. Information from the same article which you cited).

“abnormalities, neurodevelopmental phenotypes, optic atrophy, and retinal degeneration, and optic atrophy 9 (OPA9), characterized by isolated ophthalmologic phenotypes including optic atrophy and low vision[44].” Self-repeating, please rephrase.

A short section with the study limitations and future perspective would strengthen this article even more.

Conclusion after the second revision:

The authors have improved the article. However, I would like to recommend performing a second revision, after which the article might be recommended for publication.

Reviewer #2: The authors have responded to all concerns meticulously and improved the manuscript accordingly. The revised draft is improved significantly. I do not have further comments.

7. PLOS authors have the option to publish the peer review history of their article (what does this mean?). If published, this will include your full peer review and any attached files.

Reviewer #1: No

Reviewer #2: No

---

## [Author Response · Author response to Decision Letter 1]

5 Jan 2023

Response to Reviews

First of all, I would like to thank the editor and the reviewers for their recognition of my manuscript, and I have revised the manuscript accordingly, and here are the answers to the questions raised by the reviewers.

Response to Reviewer #1:.

Question 7. Discussion: “SHMT1 gene hypermethylation causes impaired folate metabolism and abnormal homocysteine (Hcy) remethylation through reduced SHMT1 expression, resulting in Hcy accumulation in the blood and hyperhomocysteinemia, which enhances oxidative stress in the kidney and induces structural damage and apoptosis in the pedal cells, causing kidney injury”. // I guess the authors mean podocytes. Answer 7. Thank you, we are indeed talking about podocytes.

Please, use podocytes instead of pedal cells.

Answer Thank you for your careful review., we have revised in the revision.

I would like to leave a few additional comments:

“A rat AKI model was constructed, and the mRNA and protein expression levels of the central genes in the model and normal groups were verified by PCR and immunohistochemistry experiments”. Please, use word control instead of normal (in this sentence and further in the manuscript).

Answer Thank you very much for your comments, we have revised in the revision.

“Screening potential core genes related to AKI by WGCNA; to explore the potential biomolecules of AKI and provide a reference for the diagnosis and treatment of AKI.” // Difficult for understanding sentence, please rephrase it.

Answer Thank you very much for your comments, we have revised in the revision.

“…genes to correlate modules with each other (using external sample signature) methods), and to…”. // Please, remove extra bracket.

Answer Thank you for your careful review, we have revised in the revision.

“… we selected modules that met a p-value less than 0.001…” // Please, start the sentence with a capital letter.

Answer Thank you for your careful review, we have revised in the revision.

“antigen repair”. The right term for this procedure is retrieval. Please also specify which buffer was used for this procedure.

Answer Thank you for your careful review, we have revised in the revision.

“Normal rats and AKI model rats were collected…”// Please exchange to “Kidney tissue from control and AKI model rats…”.

Answer Thank you very much for your comments, we have revised in the revision.

In the discussion section you are talking about SDMA and renal dysfunction. Could you please briefly specify the mechanism? (e.g. Competition for L-arginine transport. Information from the same article which you cited).

Answer Thank you very much for your comments, we have revised in the revision.

“abnormalities, neurodevelopmental phenotypes, optic atrophy, and retinal degeneration, and optic atrophy 9 (OPA9), characterized by isolated ophthalmologic phenotypes including optic atrophy and low vision[44].” Self-repeating, please rephrase.

Answer Thank you for your careful review, we have revised in the revision.

Response to Reviewer #2:.

Thank you for your recognition of our manuscript.

---

## [Decision Letter · Decision Letter 2]

24 Jan 2023

Identification of AGXT2, SHMT1, and ACO2 as important biomarkers of acute kidney injury  by WGCNA

PONE-D-22-24410R2

Dear Dr. qin,

We’re pleased to inform you that your manuscript has been judged scientifically suitable for publication and will be formally accepted for publication once it meets all outstanding technical requirements.

Kind regards,

Utpal Sen, Ph.D.

Academic Editor

PLOS ONE

Additional Editor Comments (optional):

Reviewers' comments:

Reviewer's Responses to Questions

**Comments to the Author**

1. If the authors have adequately addressed your comments raised in a previous round of review and you feel that this manuscript is now acceptable for publication, you may indicate that here to bypass the “Comments to the Author” section, enter your conflict of interest statement in the “Confidential to Editor” section, and submit your "Accept" recommendation.

Reviewer #1: All comments have been addressed

Reviewer #2: All comments have been addressed

2. Is the manuscript technically sound, and do the data support the conclusions?

Reviewer #1: Yes

Reviewer #2: Yes

3. Has the statistical analysis been performed appropriately and rigorously? 

Reviewer #1: Yes

Reviewer #2: Yes

4. Have the authors made all data underlying the findings in their manuscript fully available?

Reviewer #1: Yes

Reviewer #2: Yes

5. Is the manuscript presented in an intelligible fashion and written in standard English?

Reviewer #1: Yes

Reviewer #2: Yes

6. Review Comments to the Author

Reviewer #1: I would like to thank the authors for considering the comments from the previous round of revisions. All the recommendations have been addressed in the article. My conclusion: the manuscript by Wei and co-authors is recommended for publication.

Reviewer #2: The authors have responded to all concerns meticulously and improved the manuscript accordingly. The revised draft is improved significantly. I do not have further comments. I recommend the revised draft for publication.

7. PLOS authors have the option to publish the peer review history of their article (what does this mean?). If published, this will include your full peer review and any attached files.

Reviewer #1: No

Reviewer #2: No

---

## [Editor Report · Acceptance letter]

26 Jan 2023

PONE-D-22-24410R2 

Identification of AGXT2, SHMT1, and ACO2 as important biomarkers of acute kidney injury  by WGCNA 

Dear Dr. qin:

I'm pleased to inform you that your manuscript has been deemed suitable for publication in PLOS ONE. Congratulations! Your manuscript is now with our production department. 

Kind regards, 

on behalf of

Dr. Utpal Sen 

Academic Editor

PLOS ONE